# Efficacy of Direct or Indirect Use of Probiotics for the Improvement of Maternal Depression during Pregnancy and in the Postnatal Period: A Systematic Review and Meta-Analysis

**DOI:** 10.3390/healthcare10060970

**Published:** 2022-05-24

**Authors:** Klavdija Čuček Trifkovič, Dušanka Mičetić-Turk, Sergej Kmetec, Maja Strauss, Hannah G. Dahlen, Jann P. Foster, Sabina Fijan

**Affiliations:** 1Faculty of Health Sciences, University of Maribor, Žitna ulica 15, 2000 Maribor, Slovenia; klavdija.cucek@um.si (K.Č.T.); sergej.kmetec1@um.si (S.K.); maja.strauss@um.si (M.S.); 2Faculty of Medicine, University of Maribor, Taborska ulica 8, 2000 Maribor, Slovenia; dusanka.micetic@um.si; 3School of Nursing and Midwifery, University of Western Sydney, Parramatta, NSW 2751, Australia; h.dahlen@westernsydney.edu.au (H.G.D.); j.foster@westernsydney.edu.au (J.P.F.); 4Ingham Research Institute, Liverpool, NSW 2170, Australia; 5NSW Centre for Evidence Based Health Care: A JBI Affiliated Group, Parramatta, NSW 2751, Australia

**Keywords:** probiotics, pregnancy, postpartum, depression, prevention, psychobiotics

## Abstract

The mother and infant form a unique bond, with maternal mental health affecting the interactions with the infant and infant behaviours impacting maternal mental health. One of the possible mechanisms influencing maternal mental health is the manipulation of the gut-brain axis by consuming probiotic supplements. Probiotics can also have an indirect influence on maternal mental health via the modulation of the infant microbiome and consequently improving the infant’s health and thus, indirectly leading to an improvement in maternal mood. This systematic review evaluated the efficacy of probiotics on maternal mental health by searching for randomised controlled trials via international databases: Cochrane Library, PubMed, Scopus, ScienceDirect, and Web of Science until January 2022. A meta-analysis was performed using the Cochrane Collaboration methodology where possible. We found seven clinical trials that included the word probiotics and addressed maternal depression and/or anxiety. Of these, five trials investigated the influence of maternal probiotic supplementation on the gut-brain axis. Two trials investigated the indirect influence of probiotics on maternal depression via supplementation of probiotics by infants and subsequent influence on the crying of colicky infants. Meta-analysis of two studies of pregnant and postnatal women and two studies of infants consuming probiotics on the outcome of the Edinburgh Postnatal Depression Scale for mothers showed no statistical difference. The findings indicate that maternal depression is very complex and is influenced by various bidirectional factors. One of the factors that can improve maternal mental health is probiotics, however, careful consideration must be given to correct strain selection as strain-specific effectiveness was observed. Further well-designed, robust clinical studies are warranted.

## 1. Introduction

According to the World Health Organization (WHO), mental health is an integral part of health and well-being, which is described as: “a state of complete physical, mental and social well-being and not merely the absence of disease or infirmity” [1]. Anxiety is defined as “an emotion characterized by feelings of tension, worried thoughts and physical changes, such as increased blood pressure” according to the American Psychological Association. The WHO defines anxiety as a mental disorder characterised by feelings of anxiety and fear and depressive disorders are described as mental disorders characterised by sadness, loss of interest or pleasure, feelings of guilt or low self-worth, disturbed sleep or appetite, feelings of tiredness, and poor concentration. Depression can be long-lasting or recurrent, substantially impairing an individual’s ability to function at work or school or cope with daily life. At its most severe, depression can lead to suicide [2].

Increasing research into the microbiota-gut-brain axis may provide a new avenue of investigation to treat anxiety, depression, and cognition [3,4,5,6,7]. A range of potential mechanisms by which the gut microbiota affects central nervous system function have been proposed. These include altered microbial composition, immune inactivation, vagal nerve activation, tryptophan metabolism, gut hormone response, and through production of microbial neuroactive substances, bacterial cell wall sugars, or other metabolites [3,4,5,7]. There is evidence that less diversity of the intestinal microbial community is associated with neuropsychiatric disorders, including depression and anxiety in mothers and offspring. Assessing the maternal and child’s microbial communities may be an important missing component in mother-infant attachment-based therapies during the treatment of perinatal depression [8]. One approach to improving maternal mental health in the perinatal period is targeting the gut-brain axis by modulating the gut microbiota [9].

The administration of probiotics and prebiotics can achieve the microbiota-gut-brain axis homeostasis. According to the consensus statement by the International Scientific Association for Probiotics and Prebiotics, probiotics are defined as “live microorganisms that, when administered in adequate amounts, confer a health benefit on the host” [10]. Prebiotics are defined as a substrate selectively utilised by host microorganisms conferring a health benefit [11]. The possible applications of probiotics based on well-designed clinical trials and meta-analyses are numerous, including treatment or prevention of several disorders, such as: antibiotic and *Clostridium difficile*-associated diarrhoea, irritable bowel syndrome and inflammatory bowel disease, to anxiety, wound healing, and even depression [12,13,14,15,16,17]. The influence of probiotics on the gut-brain axis has been investigated in several recent clinical trials with positive results, including the improvement of psychiatric and gut-barrier-associated traits [18] and cognitive functions [19] in patients with a major depressive disorder; the mitigation of anxiety symptoms in patients with anxiety disorders [20,21], and the improvement of mental state, sleep quality and the gut microbiota of healthy young adults under stressful conditions [22]. These findings support the importance of the interactions between the digestive system and the brain functions via the microbiota-gut-brain axis, thus indicating the significant role of the microbiota in the modulation of the central nervous system, including affective and cognitive functions [19]. Additionally, probiotics are among nutrients that are provisionally recommended for adjunctive use in mental disorders according to the World Federation of Societies of Biological Psychiatry (WFSBP) and the Canadian Network for Mood and Anxiety Disorders (CANMAT) [23]. Some researchers refer to probiotics administered with the purpose of supporting mental health wellbeing as psychobiotics [24,25,26,27]. However, the International Scientific Association for Probiotics and Prebiotics has published a consensus statement for the terms probiotics [10], prebiotics [11], synbiotics [28], postbiotics [29] and fermented foods [30], but not for pharmabiotics, immunobiotics, psychobiotics, etc. [31].

The mother and infant form a deep bond, and their interactions can be mediated by various biological factors, such as mother and infant characteristics, including mood, depression, maternal behavior style, age, biological risk, disability, temperament, irritable infant—for example, due to colic and other health and disease conditions [32,33,34,35]. Perinatal depression is a significant public health problem due to its negative impact on maternal well-being and long-term adverse effects on children. During the prenatal period, depression can result in poorer childbirth outcomes, including pre-term delivery and low birth weight or small for gestational age infants [8,36]. In the postpartum period, maternal suicides account for 20% of all postpartum deaths, making it one of the leading causes of maternal mortality [37]. Mother-infant interaction and maternal responsiveness and sensitivity and the rapidly developing infant brain are hypothesised mechanisms by which perinatal depression affects child development [8]. Postpartum depression creates an environment that is not conducive to the personal development of mothers or the optimal development of a child; therefore, it is important to detect and treat depression during the postnatal period as early as possible to avoid harmful consequences [38].

Published clinical trials have shown that certain probiotic strains are beneficial for either perinatal and postnatal maternal health or women’s health generally, including reducing gestational diabetes mellitus [39] mastitis [40,41], bacterial or yeast vaginosis [42,43,44] as well as improving vaginal health [45], allergic scores [46], immune factors [47] and the mental health-depression-anxiety symptoms [48,49] of mothers. There are also several ongoing clinical trials on maternal health and probiotics with published study protocols [50,51,52].

A recent review by Desai and co-authors [9] that included three randomised controlled trials investigated the effectiveness of probiotic, prebiotic, or synbiotic supplementation during pregnancy to improve perinatal maternal health. Desai and co-authors [9] focused only on probiotic supplementation during pregnancy and found that there is limited but promising evidence regarding the effectiveness of probiotics during pregnancy to reduce anxiety symptoms and reduce the proportion of women scoring above a cut-off depression score. Our review aimed to assess the influence of probiotics on improving maternal depression and anxiety, whether taken directly by pregnant women or after birth, as well as the indirect influence of probiotics on maternal mental health if consumed by infants, as all factors can influence maternal mental health.

## 2. Materials and Methods

We performed a systematic review using the method outlined by Higgens et al. of examination, analysis, synthesis of results and compilation [53]. We chose this type of review because it is more specific and addresses a focused range of outcomes [54]. Therefore, we included randomised controlled trials (RCTs), which examined the effect of probiotics or psychobiotics on the occurrence of depression during women’s pregnancy and in the postnatal period. We followed the PRISMA guidelines [55,56]. We formulated the research according to the PICO strategy (Population, Intervention, Comparison, Outcome) compilation [53].

### 2.1. Research Question

Are probiotics or psychobiotics (I) compared to placebo (C) given to the mother or infant (P) effective in reducing maternal depression during pregnancy and the postnatal period (O)?

### 2.2. Search Strategy

The search was performed using search terms in English: probiotics, psychobiotics, pregnant, depression and a clinical trial with their synonyms and Boolean operators (AND/OR). The final search term was: (probiotic* OR psychobiotic*) AND (mother OR pregnant) AND depression AND “clinical trial”). The systematic search was carried out in the following databases: Cochrane Library, PubMed, Scopus, ScienceDirect, and Web of Science. We used the following search limits: research articles published in English, German and/or Slovenian related to the research topic and all studies published up to January 2022. Data were collected between December 2021 and January 2022. We used the same search terms, search limits, inclusion, and exclusion criteria in all the databases.

### 2.3. Eligibility Criteria

Predetermined inclusion and exclusion criteria were applied as presented in Table 1.

### 2.4. Critical Assessment

The methodology of the studies was assessed using the Joanna Briggs Institute critical appraisal tool (JBI) for randomised controlled trials [57]. Based on Camp and Legge’s [58] recommendation, we evaluated the studies as medium-high quality (70–79%); high quality (80–90%) and excellent quality (90% or more). The authors S.F., S.K. and M.S. conducted the critical assessment.

### 2.5. Data Extraction and Synthesis of Data

The authors S.F. and M.S. conducted data extraction from the included clinical studies based on the predefined data extraction criteria. Both authors checked with each other and managed any queries and differences. The author J.F. entered data into Review Manager software (RevMan). We performed statistical analyses using Cochrane’s Review Manager [59]. We analysed continuous data using mean differences (MDs) and reported the 95% CI on all estimates. We used the random-effects model for all meta-analyses and assessed the heterogeneity between the included trials, using the I^2^ statistic. The degree of heterogeneity was graded as non-existent or minimal for an I^2^ value of less than 25%, low for an I^2^ value of 25–49%, moderate for an I^2^ value of 50–74%, and high for an I^2^ value of 75–100%. One author (S.F.) checked the data of the metanalysis.

## 3. Results

The search of the included databases (PubMed, Scopus, Cochrane Library, ScienceDirect, Web of Science) identified 195 records imported into the reference management program EndNote 20. We also conducted a hand-searching of the grey literature, and we identified one record. With the help of this program, we identified and eliminated 15 duplicates. The remaining search records (*n* = 181) were imported into an Excel 2016 file, where two authors (S.K. and S.F.) independently reviewed titles and abstracts. Those unrelated to our study’s topic and that did not meet the inclusion criteria (*n* = 174) were excluded. Ninety-nine studies found by the search did not investigate the effect of probiotics on depression during pregnancy or in the postnatal period and 53 studies did not display the effect of probiotics on the occurrence of depression during pregnancy or in the postnatal period. Twelve studies included participants, other than pregnant women or women during the postnatal period, three studies were not in English, Slovene, or German and seven papers were not clinical trials.

The authors reviewed the full-text papers for suitability (*n* = 7). Seven articles were included in the final analysis and synthesis [48,49,60,61,62,63,64]. Studies were conducted in Australia, New Zealand, China, the Netherlands, Finland, and Iran. One thousand two hundred fifty-six participants completed mental health evaluations using various tools. All studies were placebo-controlled and blinded. The selection of the studies was conducted according to the PRISMA statement and is presented in Figure 1 [56,65].

### 3.1. Results of Critical Assessment

The clinical trials using the Joanna Briggs Institute critical appraisal tool for randomised controlled trials [57] are noted in Table 2. The clinical trial by Browne [60] and co-authors was conducted in the Netherlands and they have a previously published paper with full details about procedures, methods and outcome measures [66]. The clinical trial conducted by Dawe and co-authors [61] was a secondary analysis pertaining to antenatal mental health outcomes of the Healthy Mums and babies (HUMBA) trial with other data from the trial published elsewhere [67,68]. The clinical trial by Hulkkonen and co-authors [49] did not contain information regarding the randomisation process, concealment of allocation to treatment groups, or blinding of outcome assessors; however, another publication [69] form the same clinical trial contained this information. The clinical trial by Slykerman and co-authors [48] was part of the Probiotics in Pregnancy Study (PIP Study). It has a previously published protocol [3] and an additional paper published from the same clinical trial focusing on the influence of probiotics on gestational diabetes [70]. Details of the clinical trial by Sung and co-authors [64] are described in the published protocol [71].

The clinical trial by Mi and co-authors [62] was single-blinded. The parents were blinded, but no information was given on blinding to treatment assignment (question 5), or if outcome assessors were blind to treatment assignment (question 6). Therefore, all two critical appraisal questions were rated as unclear. Mirghafourvand and co-authors [63] described their study as a triple-blinded trial, as those involved in the sampling and data collection, the analysers, and the participants were unaware of the type of intervention received.

Question 3 (similar baseline characteristics at baseline) was rated unclear for Hulkkonen [49] because they did not divide their reported clinical characteristics separately for each treatment group. Question 11 was rated unclear for four studies [48,61,62,64] as either no information in the studies is given on the number of raters for statistical analysis in the text or no information on author contributions is given, or information is given that only one author conducted all statistical analysis.

All studies except the clinical trial by Mi and co-authors [62] were rated as superior quality according to Camp and Legge’s recommendation [58], as the percentage was 92% or 100%, whilst the study by Mi and co-authors [62] was 77% since it was single-blinded and three questions of the Joanna Briggs Institute critical appraisal tool assessed blinding.

### 3.2. Characteristics of Reviewed Articles

The characteristics of the reviewed articles are described in Table 3.

Two studies [62,64] did not assess improvement of maternal depression via the gut-brain axis, but as an indirect consequence of administration of probiotics (in both studies, *Limosilactobacillus reuteri* DSM 17938 was used) to babies with the intent to reduce infantile colic. In the study by Mi and co-authors [62], *Limosilactobacillus reuteri* DSM 17938 was a probiotic given to infants for 28 days. Throughout the study period, parents’ satisfaction was higher and improvement in maternal depression was observed as lower EPDS (Edinburgh Postnatal Depression Scale) scores in the probiotic group. On the other hand, although the clinical study by Sung and co-workers [64] assessed the same probiotic strain *Limosilactobacillus reuteri* DSM 17938 for effectiveness in reducing infant colic, no difference was observed in maternal mental health with the EPDS score. Additionally, no difference was observed for reduced colic, which could account for the indirect lack of difference in the EPDS score for maternal mental health.

Of the remaining five studies noted in Table 3, two found improved mental health outcomes [48,49]. The clinical study by Hulkkonen and co-authors [49] found a modest impact on the depressive symptoms in pregnant and postpartum women after supplementation with fish oil and *Lacticaseibacillus rhamnosus* HN001 and *Bifidobacterium animalis* subsp. *lactis* 420. This study focused on overweight women from early pregnancy until six months postpartum. The depressive and anxiety symptoms were assessed with the Edinburgh Postnatal Depression Scale (EPDS) and Anxiety subscale of Symptoms Checklist (SCL-90) at early and late pregnancy and three, six and 12 months postpartum. A statistically significant intervention effect was seen during pregnancy. The probiotics and fish oil group had lower EPDS scores at 12 months postpartum than the placebo group; however, they were not statistically significant. Therefore, the intervention had a modest impact on depressive symptoms. The diet, as well as obstetric events, also impacted depressive and anxiety symptoms. The study by Slykerman and co-authors [48] also used the strain *Lacticaseibacillus rhamnosus* HN001 and found that supplementation resulted in significantly lower postnatal anxiety scores. The probiotic was given during pregnancy and postpartum for up to 6 months. The EPDS was also used to assess maternal mood, and a six-month follow-up was conducted. This study demonstrated a significantly lower prevalence of symptoms of depression and anxiety postpartum in women supplemented with the probiotic *Lacticaseibacillus rhamnosus* HN001 during and after pregnancy than in those given a placebo. Furthermore, the number of women reporting clinically significant anxiety levels on screening was significantly lower in the probiotic group. Barthow and co-authors published the protocol for this study [3]. The primary outcomes were the development and severity of eczema and atopic sensitisation [72] in infants and a reduction in maternal gestational diabetes mellitus [70].

The last three clinical studies by Browne [60] Dawe [61], and Mirghafourvand [63] included pregnant women only. All three clinical studies focused on other outcomes and the mental health of pregnant women was a secondary outcome. In the pilot trial [60] no improvement in the mood of pregnant women after supplementation with a probiotic mixture (*Bifidobacterium bifidum* W23, *Bifidobacterium lactis* W51, *Bifidobacterium lactis* W52, *Lactobacillus acidophilus* W37, *Levilactobacillus brevis* W63, *Lacticaseibacillus casei* W56, *Ligilactobacillus salivarius* W24, *Lactococcus lactis* W19 and *Lactococcus lactis* W58) was found. There were also no differences in the mental health outcomes in obese pregnant women after supplementation with *Lacticaseibacillus rhamnosus* GG and *Bifidobacterium animalis* subsp. *lactis* Bb-12 [61] and no difference in the mental health of pregnant women after supplementation with conventional yoghurt containing two probiotics: *Lactobacillus acidophilus* La-5 and *Bifidobacterium animalis* subsp. *lactis* Bb-12 [63].

### 3.3. Outcome of EPDS Depression Score for Pregnant or Postnatal Women

A meta-analysis of two trials [48,49] involving a total of 512 women consuming probiotics during pregnancy and up to 6 months after birth was conducted to assess the outcome of EDPS depression scores at 12-months postpartum (Figure 2). The results showed no statistically significant difference between the probiotic and placebo groups (MD −0.35, 95% CI −2.29 to 1.58; I^2^ = 78%). However, the results show high heterogeneity (I^2^ = 78%) as the study by Slykerman and co-authors [48] included women with healthy pregnancies, whilst the study by Hulkkonen et al. and co-authors [49] included overweight pregnant women.

The meta-analysis for the outcome of the EPDS depression scores by Browne and co-authors [60], Dawe and co-authors [61] and Mirghafourvand and co-authors [63] of pregnant women only was not possible due to different methods of reporting the outcomes. Browne and co-authors [60] observed no effect between the probiotic and placebo groups [F(2,38) = 0.8, *p* > 0.05] for the EPDS depression scores. Dawe and co-authors [61] assessed mental health outcomes at 36 weeks of pregnancy and found no effect between the probiotic and placebo group (*p* = 0.527) on the outcome of the EPDS depression scores. Mirghafourvand and co-authors [63] did not use the EPDS score; they used the SF-36 questionnaire which assessed the mental aspects of quality of life but found no differences between both groups.

The meta-analysis for the outcome of the EPDS depression scores of the mothers of infants consuming probiotics in two studies by Mi and co-authors [62] and Sung and co-workers [64] was also conducted (Figure 3). A total of 193 mothers were assessed after four weeks of supplementation of infants in order to treat infant colic, but no statistical difference was found between both groups MD −0.92, 95% CI −3.64 to 1.80; I^2^ = 72%). In both studies the infants received the same probiotic strain *Limosilactobacillus reuteri* DSM 17938 for one month; however, the infants in the Australian study by Sung and co-authors [64] were slightly older (mean age 7.5 weeks at beginning of the study) than the infants in the Chinese study (mean age 29.7 days) by Mi and co-authors [62].

No other meta-analysis could be undertaken due to different measurement tools used and/or methods of reporting the outcomes.

## 4. Discussion

This systematic review focused on the efficacy of administering probiotics during pregnancy and in the postnatal period to improve mothers’ mental health. Our meta-analysis did not clearly prove an overall beneficial effect of the chosen probiotics, however, certain strains proved to have some impact on improving maternal mental health. Furthermore, certain probiotic strains that were consumed directly by the pregnant or postnatal women exhibited a slight impact on reducing depression. The review also showed that probiotic supplementation of infants leads to the improvement of colicky symptoms and indirectly influences the reduction of postnatal depression symptoms.

A similar systematic review and meta-analysis by Desai and co-authors [9] focused on the effectiveness of probiotic, prebiotic and synbiotic administration on perinatal maternal mental health; however, it focused only on supplementation during pregnancy. It is well known that probiotic supplementation of mothers during breastfeeding or direct supplementation to the infant has several benefits for the infant, including developing a healthy microbiome, reduction in colic and regurgitation, the development of a healthy immune system, and reduction in the risk of atopic eczema [73,74,75]. Several studies also found positive effects of probiotic supplementation for mothers, including preventing or treating mastitis [40], improving immunological factors [47] as well as reducing vaginosis and improving general vaginal health [42,45]. Poor maternal mental health has been associated with various pregnancy and child health complications. On the other hand, a recent review [76] warned that there could be an increased risk of pre-eclampsia with probiotic administration for the prevention of gestational diabetes, therefore, caution is necessary.

A systematic review and meta-analysis by Zhang and co-authors of non-pregnant healthy adults concluded that current evidence suggests that probiotics can reduce emotional stress levels in healthy volunteers and alleviate stress-related subthreshold anxiety/depression levels [77]. Psychological distress is a very real problem for new mothers and depressive symptoms and anxiety stress can potentially affect maternal mood and bonding to the offspring [60]; therefore, it could be beneficial for mothers taking probiotics during pregnancy to continue with probiotics supplementation after birth. Whilst probiotics are not intended to be a single management option for psychological distress, they can be an important supportive therapy.

According to the consensus statement of the International Probiotic and Prebiotic association [17], several probiotics have strain-specific health benefits. This was also obvious from the investigated clinical studies where one strain: *Lacticaseibacillus rhamnosus* HN001, used in the clinical studies by Hulkkonen et al. and Slykerman et al. [48,49] exhibited a beneficial effect on mental health, whilst another strain of the same species, namely *Lacticaseibacillus rhamnosus* GG used in the study by Dawe et al. [61] did not exhibit any beneficial effect on mental health compared to the placebo group.

Consideration of the possible synergistic effects of probiotic mixtures is also important. However, in our review, the beneficial effect of the *Lacticaseibacillus rhamnosus* HN001 strain on mental health was not greater when combined with *Bifidobacterium animalis* subsp. *lactis* 420 [49] compared to supplementation with only the HN001 strain [48]. Health specific benefits were also found for the two-strain probiotic containing *Lactobacillus acidophilus* La5 and *Bifidobacterium animalis* subsp. *lactis* Bb-12, used in the study by Mirghafourvand and co-authors [63]. Although this strain combination did not provide an improvement in mental health during pregnancy, reduced constipation was observed thus proving that this probiotic combination did indeed exhibit a health benefit, just not on maternal mental health.

It is known that infantile colic is associated with higher maternal depression symptoms scores and lower quality of life scores [78,79]. Two clinical studies included in our review by Mi and co-authors and Sung and co-authors [62,64] used the same strain of *Limosilactobacillus reuteri* DSM 17938, but statistically significant improvement in maternal mental health symptoms was found only in the study by Mi and co-authors [62]. Infantile colic, like regurgitation and subsequent infant crying, is very stressful for parents. However, these two studies investigated maternal depression based on distress due to infant colic. Therefore, the influence on maternal stress was indirect, indicating that maternal depression is tightly linked with infant well-being. Although both studies observed the same primary outcome using the same probiotic strain (*Limosilactobacillus reuteri* DSM 17938), the study conducted by Mi and co-authors was conducted on infants from China with a positive outcome for reduced colic and reduced maternal depression in the probiotic group, whilst in the Australian study by Sung and co-authors no statistically significant difference was found for reduction of colic or for maternal mental health. In the Chinese study, the average age of the infants enrolled in the study was 29.7 days for the probiotic group and 28.6 days for the placebo group, whilst in the Australian study the average age of the infants entering the study was 7.5 weeks for the probiotic group and 8.9 weeks for the placebo group, thus these infants were older and perhaps the opportunity for the probiotics to reduce infantile colic was distinct; thus leading to the difference in the consequent effect on EPDS depression score of mothers. The authors of the Australian study [64] also emphasised that they recruited infants from the ‘emergency or urgent care settings’ and so were already exhibiting symptoms of crying as the most common age for development of colic is about 3 to 6 weeks of age, with most resolved by three months of age [79]. Therefore, timely administration of probiotics to breastfeeding mothers or directly to infants, such as *Limosilactobacillus reuteri* DSM 17938 [80] could reduce infantile colic and thus benefit the infant and also indirectly influence maternal mental health. In the case of probiotic supplementation directly to infants, this benefit is not via the gut-brain axis of the mother. With mothers with mental health disorders found to be nearly five times as likely to have a baby admitted with GOR/GORD in the first year after birth [81] this is an important consideration. It is well known that the mode of birth, as well as breastfeeding, also influences the infant microbiota. Infants born via caesarian section have different gut microbiota compared to vaginally born infants. Infants born vaginally and fully breastfed infants have gut microbiota dominated by Bifidobacteria [82]. Thus, infant probiotic supplementation could be warranted when babies are born by caesarian section or when not exclusively breastfeeding.

Improvement of maternal mental health was observed in only three [48,49,62] of the seven clinical studies included in our review; the studies by Dawe and co-authors and Browne and co-authors [60,61] both investigated the influence of supplementation with probiotics only up to delivery and observed no beneficial effects on mental health, anxiety or depression. This could be due to two reasons, inappropriate strain selection, as not all probiotic strains influence mental health, as well as the impending birth and the natural elevation of anxiety connected with this [83]. However, since some of the clinical trials exhibited a positive effect of probiotics on reducing postnatal depression, the clinical environment can promote probiotics for reducing symptoms of postnatal depression.

## 5. Conclusions

Maternal mental health is very complex and can be influenced by the gut-brain axis or by other external influences, such as infant mood. Current evidence has shown that certain probiotic strains are better than placebo in exhibiting a positive health benefit for maternal mental health, based on individual clinical trials. The analysis of the clinical studies shows that strain selection is one of the most decisive factors in showing a statistically significant difference compared to placebo. The strain *Lacticaseibacillus rhamnosus* HN001 was effective in reducing maternal depressive symptoms when maternal supplementation was continued in the postnatal period; the supplementation of the probiotic *Limosilactobacillus reuteri* DSM 17938 to the infants was effective in reducing colic, and therefore, indirectly reducing maternal depressive symptoms if the probiotic was given to the infant before colicky symptoms peaked. However, more robust, well-designed clinical trials are needed. Future studies should include probiotic supplementation after birth and careful consideration should be given to measure outcomes in a reliable way. Accurate reporting of allocation concealment and blinding of participants and those delivering the treatment or assessing the outcomes is important in future trials. Importantly clinical trials should not be influenced by probiotic manufacturer funding.

## Figures and Tables

**Figure 1 healthcare-10-00970-f001:**
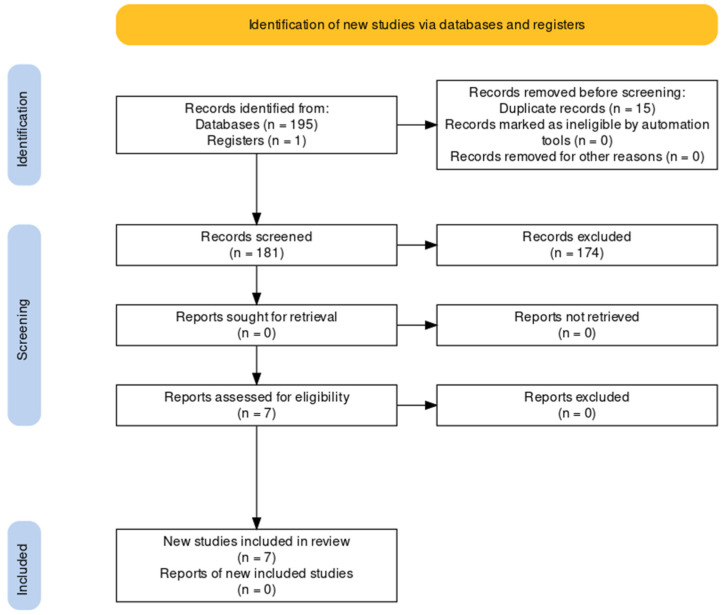
PRISMA flow diagram of study selection.

**Figure 2 healthcare-10-00970-f002:**
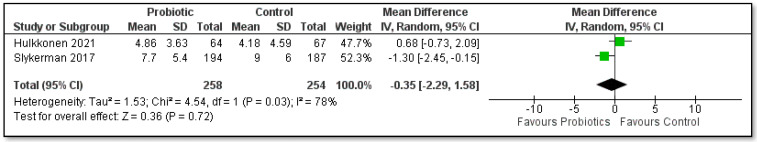
Meta-analysis of the effect of probiotic supplementation for pregnant women and postnatal women on EPDS depression score.

**Figure 3 healthcare-10-00970-f003:**
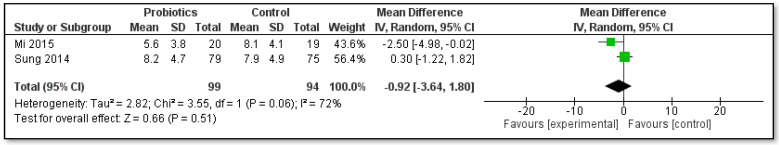
Meta-analysis of the effect of probiotic supplementation of colicky infants on the EPDS depression score of mothers.

**Table 1 healthcare-10-00970-t001:** Research strategy containing inclusion criteria, exclusion criteria and search limits.

Database:	PubMed, Cochrane Library, ScienceDirect, Scopus in Web of Science	
Inclusion criteria		Exclusion criteria
Participants:	Pregnant women or postpartum women over 18 years of age (or infants if they indirectly affected maternal mental health).	Pregnant women under 18 years of age. Infants, if maternal mental health was not assessed.
Intervention /treatment:	Effect of probiotics (or psychobiotics) on the occurrence of depression in women during pregnancy and in the postnatal period.	The effect of probiotics (or psychobiotics) on depression during the woman’s pregnancy and in the postnatal period is not included.
Comparison	Studies included a comparator or control group.	Studies without a control group.
Outcome:	The article displays the direct or indirect effect of probiotics (or psychobiotics) on the occurrence of depression during pregnancy or in the postnatal period.	The article does not display the direct or indirect effect of probiotics (or psychobiotics) on the occurrence of depression during pregnancy or in the postnatal period.
Types of research:	Randomised controlled trials.	Research articles (quantitative, qualitative, and mixed methods research). Systematic review articles or other types of reviews. Duplicates, commentaries, editorials, conferences, and research protocols.
**Search limits**		
Timeframe:	Searches were performed from inception to January 2022.	
Language:	English, German, Slovenian.	

**Table 2 healthcare-10-00970-t002:** Quality assessment checklist of included clinical trials using the Joanna Briggs Institute critical appraisal tool for randomised controlled trials.

First Author, Year (In Alphabetical Order)	1	2	3	4	5	6	7	8	9	10	11	12	13	Study Quality
Browne, 2021 [60]	✓	✓	✓	✓	✓	✓	✓	✓	✓	✓	✓	✓	✓	superior
Dawe, 2020 [61]	✓	✓	✓	✓	✓	✓	✓	✓	✓	✓	◯	✓	✓	superior
Mirghafourvand, 2016 [63]	✓	✓	✓	✓	✓	✓	✓	✓	✓	✓	✓	✓	✓	superior
Mi, 2015 [62]	✓	✓	✓	✓	◯	◯	✓	✓	✓	✓	◯	✓	✓	medium high
Hulkkonen, 2021 [49]	✓	✓	◯	✓	✓	✓	✓	✓	✓	✓	✓	✓	✓	superior
Slykerman, 2017 [48]	✓	✓	✓	✓	✓	✓	✓	✓	✓	✓	◯	✓	✓	superior
Sung, 2014 [64]	✓	✓	✓	✓	✓	✓	✓	✓	✓	✓	◯	✓	✓	superior

Note possible answers: **✓**—YES, **✕**—NO, **◯**—Unclear, not applicable (NA); 1. Was true randomisation used for assignment of participants to treatment groups? 2. Was allocation to treatment groups concealed? 3. Were treatment groups similar at the baseline? 4. Were participants blind to treatment assignment? 5. Were those delivering treatment blind to treatment assignment? 6. Were outcomes assessors blind to treatment assignment? 7. Were treatment groups treated identically other than the intervention of interest? 8. Was follow up complete and if not, were differences between groups in terms of their follow up adequately described and analysed? 9. Were participants analysed in the groups to which they were randomised? 10. Were outcomes measured in the same way for treatment groups? 11. Were outcomes measured in a reliable way? 12. Was appropriate statistical analysis used? 13. Was the trial design appropriate, and were any deviations from the standard RCT design (individual randomisation, parallel groups) accounted for in the conduct and analysis of the trial? Study quality rated according to the Camp and Legge’s recommendation [58].

**Table 3 healthcare-10-00970-t003:** Characteristics and description of the seven clinical studies on the influence of probiotics on maternal depression and/or anxiety.

Reference (First Author, Year)	Study Design	Population that Completed Trial Regarding Influence of Probiotics on Maternal Mental Health	Intervention				Related to Depression/Anxiety
			Active	Control	Duration	Validated Questionnaires	Main Findings
**Population consuming probiotics: pregnant women and postnatal women**							
Hulkkonen, 2021 [49], Finland	Randomised, double-blind, placebo-controlled with four intervention groups	438 pregnant overweight women at less than 18 weeks of gestation (219 in the probiotic group, 219 in the placebo group).	One capsule per day containing *Lacticaseibacillus rhamnosus* HN001 and *Bifidobacterium animalis* subsp. *lactis* 420 (2 × 10^10^ cfu). Additionally, two fish oil capsules per day.	Microcrystalline cellulose	From the first study visit up to 6 months after birth	EPDS, DEP, ANH, ANX, SCL-90.	The intervention modestly impacted depressive symptoms at follow-up (*p* = 0.039). Diet, obstetric events, and mother-reported baby colic was associated with depressive and anxiety symptoms.
Slykerman, 2017 [48], New Zealand	Randomised, double-blind, placebo-controlled trial	380 pregnant women were recruited between 14- and 16-weeks’ gestation (193 in probiotic group, 187 placebo group).	One capsule per day containing *Lacticaseibacillus rhamnosus* HN001 (10^9^ cfu).	Corn-derived maltodextrin	From 14–16 weeks gestation until birth and from birth until six months post-birth whilst breastfeeding.	EPDS, STAI-6	Women in the intervention group (HN001) had significantly lower depression (*p* = 0.037) and anxiety scores (*p* = 0.014) in the postpartum period.
**Population consuming probiotics: pregnant women only**							
Browne, 2021 [60], Netherlands	A double-blind, randomised pilot trial	40 healthy pregnant women (≥18 years) (≥26 and ≤30 weeks gestation) with elevated symptoms of depression or anxiety (20 in each group).	One powder sachet per day containing *Bifidobacterium bifidum* W23, *Bifidobacterium lactis* W51, *Bifidobacterium lactis* W52, *Lactobacillus* *acidophilus* W37, *Levilactobacillus brevis* W63, *Lacticaseibacillus casei* W56, *Ligilactobacillus salivarius* W24, *Lactococcus lactis* W19 and *Lactococcus lactis* W58 (2.5 × 10^9^ cfu).	Indistinguishable regarding colour, taste, and smell from probiotic	Taken for up to 8 weeks during pregnancy. Follow-up 4 weeks postpartum	EPDS, LEIDS-R, PRAQ-R, STAI-S, PES, APL, MAAS, MPAS, PSQI	There were no differences between the groups for improving maternal mood during gestation (*p* > 0.05). However, a significant decrease in depressive symptoms between baseline and postpartum (*p* = 0.04) was observed in both groups.
Dawe, 2020 [61], New Zealand	A randomised controlled trial	164 pregnant women with obesity; forming analytic sample between 12–17 weeks gestation (88 in probiotic group, 76 placebo group).	One capsule per day containing *Lacticaseibacillus rhamnosus* GG and *Bifidobacterium animalis* subsp. *lactis* BB12 (6.5 × 10^9^ cfu).	Microcrystalline cellulose and dextrose anhydrate	From recruitment up to 36 weeks of pregnancy	EPDS, STAI-6, SF-12-v2	Anxiety and physical well-being scores worsened over time irrespective of group allocation, and mental well-being scores did not differ between the two groups at 36 weeks (*p* > 0.05).
Mirghafourvand, 2016 [63], Iran	Triple-blind placebo randomised controlled trial	60 pregnant constipated women >18 years, gestational age between 24 and 28 weeks (30 in each group).	300 g of yoghurt three times a day containing 4.8 × 10^10^ cfu of *Bifidobacterium animalis* subsp. *lactis* Bb-12 and *Lactobacillus acidophilus* La-5.	Conventional yoghurt	Four weeks after recruitment	SF-36	There was no statistically significant difference between the groups in the mean scores of physical (*p* = 0.726) and mental (*p* = 0.678) aspects of quality of life after the intervention.
**Population consuming probiotics: infants**							
Mi, 2015 [62], China	Randomised single blind placebo-controlled study	42 infants less than four months of age, weighing between 2.5 kg and 4.00 kg and exclusively or predominantly breastfed (21 in each group).	An oil-based suspension containing *Limosilactobacillus reuteri* DSM 17938 (10^8^ cfu per day).	Identical formulation without live microorganisms	28 days	EPDS	Improvement in maternal depression was significantly higher (*p* < 0.01) in the probiotic group due to reduced crying time during infantile colic
Sung, 2014 [64], Australia	Double-blind, randomised placebo controlled trial	132 healthy breastfed or formula-fed infants less than 13 weeks of age with infantile colic (69 in the probiotic group, 63 in the placebo group).	Five drops of oil-based suspension containing *Limosilactobacillus reuteri* DSM 17938 (10^8^ cfu per day).	Maltodextrin in oil suspension	One month	EPDS, PedsQL, AQol-4R	No improvements in maternal mental health were observed (*p* = 0.44), as no differences in colic symptoms were observed (*p* > 0.05).

BMI: body mass index. EPDS: Edinburgh Postnatal Depression Scale for assessment of maternal postnatal mood. DEP: EPDS-subscale to describe non-specific depressive symptoms. ANH: EPDS-subscale to describe anhedonia; ANX: EPDS-subscale to describe anxiety symptoms. SCL-90/anxiety subscale questionnaire for assessing anxiety and other psychiatric symptoms. STAI-6: a shortened version of the original Spielberger State-Trait Anxiety Inventory for assessment of state anxiety. LEIDS-R: Leiden Index of Depression Sensitivity-Revised. PRAQ-R: Pregnancy-specific Related Anxiety Questionnaire-Revised. STAI-S: State-trait Anxiety Inventory self-report questionnaire for the assessment of general anxiety. PES: Pregnancy Experience Scale for assessing maternal appraisal of daily, pregnancy-specific hassles and uplifts. APL: Dutch Everyday Problem List. MAAS: Maternal antenatal Attachment Scale. MPAS: Maternal Postnatal Attachment Scale. PSQI: Pittsburgh Sleep Quality Index. SF-12-V2: a short form of Health survey SF-36 to assess physical and mental aspects of functional health and well-being. SF-36: quality of life (QoL) short-form health survey. AQol-4R: Assessment of quality of life. PedsQL: paediatric quality of life inventory family impact subscale to assess family functioning.

## Data Availability

Not applicable.

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
