# Peer review of "Efficacy of Direct or Indirect Use of Probiotics for the Improvement of Maternal Depression during Pregnancy and in the Postnatal Period: A Systematic Review and Meta-Analysis"

_healthcare, 2022, doi:10.3390/healthcare10060970_

Round 1

Reviewer 1 Report

Dear Authors

Thank you for the opportunity to read your manuscript, which I read with great interest.

The review manuscript is well structured, however, it needs some changes that will improve it significantly. Below you will find some points in the manuscript that need clarification, refinement, reanalysis, rewriting and/or additional information and suggestions on what can be done to improve it.

Title - The title refers to maternal depression during pregnancy and in the postnatal period and throughout the article the baby population appears, it seems to me that it should be revised.

Abstract - The abstract presents a different objective than the one presented at the end of the Introduction section, and the objective should be the same throughout the article..

Section 1 (Introduction) - this section needs some adjustments, as some information and/or points are missing or unclear, and should be included or better written (e.g. objectives of the study), I will now present some items:

What is the importance of doing this research/contribution it brings to the literature in the field?

-Why should readers be interested?

- What problem/ gap resolve/fill this research?

- How will the proposed study remedy this deficiency/lacuna/problem and provide a unique contribution to the literature.

Section 2 (Materials and Methods) - in this section some points should be clarified and improved, namely:

- Review research question, should be in line with the objective;

- Was the PICO strategy used? A systematic review requires clarification of the acronym PICO, as does the method followed (Higgens et al)...

- Is the population pregnant/postpartum women and infants? Clarify, because in the objective presented in the abstract, it does not mention as population babies?

- In Table 1, the participants include infants, however, the interventions and results only refer to women during pregnancy and postpartum ... needing some clarification ...

In item 2.4, they mention that they used the JBI methodology to assess the methodological quality of the studies, however they do not mention up to what percentage the studies are excluded, because in table 2, they exclude one study because the methodological quality of the study is less than 70%, it seems to me that the value of the exclusion should be stated...

The figure 1 PRISMA Flowchart, should be improved, which can be http://prisma-statement.org/prismastatement/flowdiagram.aspx https://estech.shinyapps.io/prisma_flowdiagram/

Section 3 (Results) - the results appear to be complete.

Section 4 (Discussion) - the discussion should be improved, it is not clear.

Section 5 (Conclusion) - the conclusion is too simple and a bit poor.

Author Response

Thank you for your review. We have to our best ability impleemented all remarks in the revised manuscript. A detailed description of the answers to each remark are in the attachment.

Reviewer 2 Report

This is a systematic review and meta-analysis evaluating the efficacy of probiotics on maternal mental health.

Title: Consider changing to “Efficacy of Probiotics for Improvement of Maternal Depression during Pregnant and Postnatal Periods: A Systematic Review and Meta-Analysis”.

Introduction: Recommend re-arranging paragraph 2 for more logical flow of arguments. Please make introduction more concise so that in the last paragraph, you can elaborate on the gap in the literature that this review addresses. Highlight the findings of the review by Desai and state why this present further research is needed. Stress - what is distinct about this paper?

Methods: Confusing that the term psychobiotic is used after stating that the ISAPP has not recognized the term. Please make sure terminology is consistent with the ideas presented in Introduction. Please specify exclusion criteria in 2.3.

Results: First paragraph can be significantly cut down, do not need a complete re-statement of Prisma Flow chart. Please make 3.2 much more concise by using summary descriptive statistics. Refrain from re-stating the main findings of each paper as this content should be moved to a new separate section for better clarity. Characteristics should help to understand the study design and sample population, which helps readers assess validity and generalizability. The results should be separate so readers can understand a synthesis of the data. In 3.3, some interpretation is stated (such as line 338). Move interpretation to Discussion section.

Discussion: Please structure paragraphs clearly with brief summary of result, its consistency with previous literature and whether the finding was expected/unexpected, and its social/political/clinical implication. Currently, paragraph two and onwards makes it difficult to draw the key point being made since the sentences alternate between prior research, this present review, and interpretive statements. In the paragraph beginning at line 377, it’s unclear what the conclusion/implication is regarding these differences.  It is expected that methodology between studies will differ but why is this important and why was this detailed so thoroughly? Please revise section for such issues with succinctness and clarity.

Conclusion: The main findings of this study are mentioned but should be highlighted more firmly. Rather than using words like “seem”, aim for methodologically rigorous words such as noting correlation.

Writing style: Writing would benefit from improving succinctness and ensuring logical flow of points. Please have clear topic sentences and ensure that large paragraphs of details have a strong conclusion explaining why that information was presented.

Author Response

(The authors gave the same response as above.)

Round 2

Reviewer 1 Report

Dear authors, thank you for the changes. The article has become clearer and I agree with the changes made.

Reviewer 2 Report

My comments have been well addressed. Thank you very much for your revisions.